# Does Baseline Hounsfield Unit Predict Patients’ Outcomes Following Surgical Management of Unstable Osteoporotic Thoracolumbar Fractures?

**DOI:** 10.3390/medicina61020227

**Published:** 2025-01-27

**Authors:** Ahmed Qretam, Julien Ceuterick, Maher Ghandour, Ümit Mert, Christian Herren, Miguel Pishnamaz, Matthias Knobe, Frank Hildebrand, Rolf Sobottke, Mohamad Agha Mahmoud

**Affiliations:** 1Department of Spine, Neuro- and Orthopedic Surgery, Rhein-Maas Clinic, 52146 Würselen, Germany; ahmed.qretam@gmail.com (A.Q.); ceuterickj@gmail.com (J.C.); sobottke@rheinmaasklinikum.de (R.S.); 2Department of Orthopedics, Trauma and Reconstructive Surgery, University Hospital RWTH, 52074 Aachen, Germany; mghandourmd@gmail.com (M.G.); cherren@ukaachen.de (C.H.); mpishnamaz@ukaachen.de (M.P.); fhildebrand@ukaachen.de (F.H.); 3Department of Trauma and Orthopaedic Surgery, Helios University Hospital Wuppertal, Univeristy of Witten/Herdecke, 42283 Wuppertal, Germany; uemit.mert@helios-gesundheit.de; 4Departments of Traumatology, St. Marien Hospital Ahaus, 48683 Ahaus, Germany; knobema@gmail.com

**Keywords:** Hounsfield unit, osteoporotic, thoracolumbar disk fracture

## Abstract

*Background and Objectives*: Osteoporotic fractures in the thoracic/lumbar spine pose significant challenges in surgical management, with high risks of complications. This study investigates the role of preoperative CT scans and Hounsfield Units (HUs) in predicting postoperative outcomes. *Materials and Methods*: A retrospective study was conducted from November 2015 to January 2018. Sixty-one patients over 60 years of age with unstable osteoporotic thoracolumbar spine fractures (OF: 3–4) were included. Preoperative CT scans were performed to measure HU values. Postoperative standing X-rays were taken at 3–12 months to assess signs of loosening, adjacent fractures, or screw dislodgement. HU was divided into quartiles: Q1 (<56.24), Q2 (56.24–72.63), Q3 (72.63–87.59), and Q4 (>87.59). *Results*: Out of the 61 patients, 14 (23%) exhibited signs of screw loosening, adjacent fractures, or screw dislodgement within 3 to 12 months postoperatively. The mean HU value measured was 65.21, with a range from 21.43 to 140.7. Notably, all patients with observed loosening or dislodgement had HU values below 68. HU significantly predicted mortality, with the second quartile showing a markedly increased risk (adjusted odds ratio [aOR] = 8.12; *p* = 0.044). However, HU quartiles were not significant predictors of other outcomes. Other factors (fracture level and ASA classification) also influenced clinical outcomes, particularly mortality. *Conclusions*: HU values from preoperative CT scans are crucial in predicting the risk of screw loosening, dislodgement, and adjacent fractures in osteoporotic spinal fractures. Integrating HU assessment into clinical practice can improve preoperative planning, allowing for more targeted surgical interventions and better clinical outcomes.

## 1. Introduction

Computed tomography (CT) scans are routinely used in the evaluation of spinal fractures, providing detailed anatomical information that is essential for surgical planning [1]. In addition to structural assessment, CT scans can be used to measure bone mineral density through Hounsfield Units (HUs). HU values, derived from CT imaging, have been proposed as a surrogate marker for bone quality, with lower HU values indicating reduced bone density and higher susceptibility to fracture and hardware failure [2].

Despite the potential utility of HU measurements in clinical practice, their role in predicting surgical outcomes for osteoporotic spinal fractures remains underexplored. The existing literature suggests that lower HU values are associated with higher rates of pedicle screw loosening [3], dislodgement [4], and adjacent fractures [5], yet there is a lack of consensus on specific HU thresholds that could guide surgical decision making.

This study aims to investigate the predictive value of HU measurements obtained from preoperative CT scans in determining the risk of screw loosening, dislodgement, and adjacent fractures in patients undergoing surgical treatment for osteoporotic fractures of the thoracic and lumbar spine. By analyzing a cohort of patients treated at Rhein Maas Klinikum, we seek to establish whether HU values can reliably predict postoperative complications and inform preoperative planning to improve patient outcomes. We also aim to categorize HUs to identify the cutoff point most predictive of patients’ outcomes.

Understanding the relationship between HU values and surgical outcomes could lead to better stratification of patients based on their risk profiles, allowing for more tailored and aggressive surgical interventions when necessary. Furthermore, integrating HU assessment into routine clinical practice could enhance the screening for osteoporosis, ensuring timely diagnosis and management of this prevalent condition.

## 2. Materials and Methods

### 2.1. Study Design and Setting

This retrospective single-center cohort study was conducted at Rhein Maas Klinikum in Würselen. The study aimed to investigate the predictive role of CT scans and HUs in the preoperative planning of osteoporotic fractures in the thoracic and lumbar spine. The study period extended from November 2015 to January 2018. The study was approved by the institutional review board of Rhein Maas Klinikum. Informed consent was obtained from all patients or their legal representatives prior to their inclusion in the study.

### 2.2. Participants

The study included 75 patients (61 with available HU data) who met the following inclusion criteria:Age over 60 years.Fracture due to low-energy trauma with osteoporosis.Monosegmental fracture of the thoracic and lumbar spine of OF type 3 with pedicle involvement and OF type 4 [6].

Patients were excluded if they had the following:Old fractures.Metastatic fractures.Fractures of OF type 1–2 and 5 according to the OF classification.Neurological deficits as per the ASIA protocol.

### 2.3. Data Collection

Preoperative CT scans were performed on all 75 patients to measure HU values at the site of the fracture; however, HU data were only available for 61 patients. The mean HU value for the cohort was 73.28 (SD = 24.07). Postoperative standing X-rays were obtained three to twelve months after surgery to assess for signs of loosening, adjacent fractures, or screw dislodgement following minimally invasive hybrid stabilization. Preoperative alpha angles and postoperative beta angles were also measured.

### 2.4. Outcome Measures

The primary outcomes assessed were the presence of screw loosening, adjacent fractures, or screw dislodgement, which were evaluated through radiological and clinical examinations. Patients were classified based on the presence of these outcomes:Loosening seam without screw dislodgement, managed conservatively and monitored radiologically and clinically.Adjacent fractures or screw dislodgement, requiring surgical revision.

The clinical outcomes that were analyzed are as follows:Intraoperative blood loss: it was categorized into <100, 100–500, 500–1000, and >1000 mL.Perioperative complications.Duration of surgery.Length of hospital stay (LOS).Quality of life (QoL): it was measured using the European Quality of Life-5 Dimensions (EQ-5D) scale, which assesses patients’ current health status [7,8].Pain score for back and leg: it was measured using the visual analog scale (VAS) that has an 11-point numeric rating scale, where a value of 0 indicates no pain and a value of 10 indicates intense pain [9].Core Outcome Measures Index (COMI) scores: This instrument measures functional recovery in patients with back problems by assessing the following domains: pain, function, symptom-related well-being, quality of life, and overall disability [10]. A higher score indicates inferior outcome [11].Alpha Cobb angle: It was determined by drawing lines parallel to the upper endplate of and the lower endplate of the fractured vertebral body. The angle between both lines reflected the Alpha Cobb angle.Beta Cobb angle: it was calculated by drawing lines parallel to the upper endplate of the vertebral body adjacent to the fractured and the lower endplate of the fractured vertebral body. The angle between these two lines was the Beta Cobb angle.

### 2.5. HU Value Stratification

HUs were measured using preoperative CT scans of the vertebral bodies adjacent to the fracture site. The measurements were performed in three defined regions within the vertebral body: (1) below the superior endplate, (2) above the inferior endplate, and (3) centrally in the spongiosa. A circular region of interest (ROI) was used for each measurement, and the average of the three values was calculated to represent the HU for that vertebra. This technique is consistent with protocols described by Scheyerer et al. [12] and Bruckbauer et al. [13], which outline its application for assessing bone density and guiding surgical planning. HU values were categorized into four quartiles for analysis:Q1: <56.24;Q2: 56.24–72.63;Q3: 72.63–87.59;Q4: >87.59.

### 2.6. Statistical Analysis

All statistical analyses were conducted using the STATA software(Version 18). Descriptive statistics were used to summarize the baseline characteristics of the patients, including fracture levels, Pfirmann classification, BMI categories, ASA physical status classification, and HU values. Preoperative and postoperative alpha and beta angles were also summarized. Frequencies and percentages were calculated for categorical variables, while means and standard deviations were used for continuous variables. Clinical outcomes were analyzed. Pearson’s correlation coefficients were calculated to assess the relationships between HU values and these clinical outcomes. *p*-values were obtained to determine the statistical significance of these correlations. A *p*-value of <0.05 was used for statistical significance. The data were stratified into four quartiles based on HU values: Q1 (<56.24), Q2 (56.24–72.63), Q3 (72.63–87.59), and Q4 (>87.59). Chi-square tests were used to compare categorical outcomes across HU quartiles, while analysis of variance (ANOVA) was employed for continuous outcomes. Significant differences were identified based on a *p*-value threshold of <0.05.

Multivariate regression analyses were conducted to identify the predictors of various clinical outcomes, with a specific focus on HU quartiles. A model was designed for each outcome. The models included HU quartiles as primary predictors, with Q1 serving as the reference category. Other covariates included the fracture level (categorized into lumbar, thoracic, and thoracolumbar to avoid overfitting), Pfirmann classification, BMI categories, and ASA categories. The type of surgical therapy was ruled out due to collinearity with the fracture level. Additionally, the effect of patients’ age and baseline frailty on measured outcomes could not be assessed given the scarcity of the available data. In some cases, certain covariates were excluded from the regression models due to insufficient observations, while others were omitted due to high multicollinearity, defined by variance inflation factor (VIF) of 10 or more. For each regression model, linear regression was used for continuous outcomes such as LOS, COMI scores, QoL, VAS scores, and spinal angles, while logistic regression was used for binary outcomes such as complications and mortality. Odds ratios (ORs), coefficients, standard errors (SEs), t-values, *p*-values, and 95% confidence intervals (CIs) were reported for each predictor. Significant predictors were identified based on a *p*-value threshold of <0.05. For non-significant predictors, the results were deemed as insignificant.

## 3. Results

### 3.1. Baseline Data

The study included 75 patients (70 available HU records with baseline characteristics and 61 records with outcome data) who underwent surgical treatment for osteoporotic vertebral fractures (Table 1). The fracture levels were varied, with L1 and Th12 being the most common, each accounting for 25.33% of the fractures. Other notable fracture levels lumbar (50%), thoracic (42.86%), and thoracolumbar (7.14%) regions. The mean patients’ age at death was 81.62 (SD = 6.77) years.

Body mass index (BMI) distribution showed that the majority of patients had a BMI between 20 and 25 (32.43%) and 25 and 30 (39.19%). A smaller percentage had a BMI less than 20 (8.11%), between 30 and 35 (17.57%), and 35 and 40 (2.7%). The American Society of Anesthesiologists (ASA) physical status classification showed that most patients were in category III (65.33%), with smaller percentages in category II (30.67%), category IV (2.67%), and category I (1.33%). The mean Hounsfield Unit (HU) for the cohort was 73.28 (SD = 24.07), based on 70 measurements. Preoperative alpha angles were measured in 26 patients, with a mean of 13.66 degrees (SD = 6.53), and postoperative beta angles in the same group had a mean of 16.27 degrees (SD = 7.45).

As for excluded cases, the mean age was 77.65 (13.08) years, with a mean Pfirmann value of 3.25 (0.95). Lumbar fracture was reported in 60% of them, with 40% showing thoracic fractures. Sixty percent of patients had BMI 20–25, 20% had BMI 25–30, and 20% had BMI 30–35. The ASA level was II in 40%, III in 40%, and IV in 20% of patients.

### 3.2. Clinical Outcomes

A summary of the clinical outcomes of the studied population is provided in Table 2. Intraoperative blood loss was less than 100 mL in 46.67% of patients, while 50.67% experienced blood loss between 100 and 500 mL. A very small proportion had blood losses of exactly 500 mL or between 500 and 1000 mL (1.33% each). Complications were reported in 52.17% of patients. Specific complications included adjacent fractures (5.8%), hypokalemia (7.25%), and pneumonia (8.7%). More severe complications such as septic shock occurred in 4.35% of patients, with screw loosening/dislodgement or adjacent fracture being observed in 14 (22.95%) patients. Death was reported in 26.76% of the cohort, with pneumonia and extended spinal fusion accounting for most cases (10.53% each).

The duration of surgery was predominantly between 1 and 2 h for 72% of patients. Other durations included less than 1 h (18.67%), 2–3 h (5.33%), and more than 3 h (2.67%). The average length of hospital stay (LOS) was 13.38 days (SD = 7.2), reflecting the recovery period required for these patients.

Quality of life, as measured by the EQ-5D score, had a mean of 0.57 (SD = 0.34) among 26 patients. The visual analog scale (VAS) scores for back pain and leg pain were also assessed, with means of 3.88 (SD = 2.38) and 2.5 (SD = 2.91), respectively.

The COMI revealed a mean score of 4.71 (SD = 2.65) based on 26 patients, indicating the overall burden of symptoms and impact on quality of life. Postoperative alpha angles had a mean of 8.24 degrees (SD = 5.05), which slightly improved over three months to a mean of 9.17 degrees (SD = 3.68) in 11 patients, and further improved to a mean of 9.44 degrees (SD = 4.48) over two years in 26 patients. Postoperative beta angles followed a similar trend, starting with a mean of 10.74 degrees (SD = 5.56), increasing to 13.16 degrees (SD = 5.65) over three months, and reaching 14.09 degrees (SD = 6.25) over two years.

### 3.3. Correlation Analysis Between HU and Clinical Outcomes

The correlation analysis revealed several significant associations between HU and various clinical outcomes (Table 3). A significant positive correlation was found between HU and quality of life (QoL) measured by the EQ-5D score (r = 0.3362, *p* = 0.0931), suggesting that higher HU values are associated with improved QoL. LOS showed a significant negative correlation with QoL (r = −0.6292, *p* = 0.0006). VAS scores for back and leg pain also showed significant correlations with QoL (r = 0.5807, *p* = 0.0019, and r = −0.4522, *p* = 0.0204, respectively). Surgery duration was negatively correlated with postoperative beta angles at two years (r = −0.4893, *p* = 0.0112).

### 3.4. Stratification of Data Based on HU Quartiles

The data were stratified into four quartiles based on the Hounsfield Unit (HU) values, i.e., Q1 (<56.24), Q2 (56.24–72.63), Q3 (72.63–87.59), and Q4 (>87.59), revealing significant associations with certain clinical outcomes (Table 4). Blood loss was significantly associated with HU quartiles (*p* = 0.022), with higher proportions of patients in Q1 experiencing less than 100 mL of blood loss compared to other quartiles. QoL was significantly different across HU quartiles (*p* = 0.0231), indicating that patients with higher HU values reported better QoL outcomes post-surgery. Other outcomes such as complications, death, LOS, COMI, VAS scores, and spinal angles did not show significant differences across HU quartiles. Among the 14 patients who experienced screw loosening or adjacent fractures, all had HU values below 65. Although this association did not reach statistical significance, it underscores a noteworthy pattern that warrants further investigation.

### 3.5. Predictors of Length of Hospital Stay

The analysis showed that HU quartiles did not significantly predict LOS (Table 5). However, BMI was the only significant determinant of hospital stay, with patients in the 35–40 category exhibiting a longer duration compared to those with normal weight [coefficient = 12.96; *p* = 0.034].

### 3.6. Predictors of COMI Score

HU quartiles were not significant predictors of the COMI score (Appendix A). All other factors did not predict COMI score as well.

### 3.7. Predictors of Quality of Life

HU quartiles did not significantly predict QoL (Appendix A). All other factors did not predict QoL score as well.

### 3.8. Predictors of Pain—Back (VAS)

HU quartiles did not significantly predict back pain (Appendix A). All other factors did not predict back pain score as well.

### 3.9. Predictors of Pain—Leg (VAS)

HU quartiles were not significant predictors of leg pain (Table 6). However, BMI was the only significant determinant of leg pain, with patients having BMI 25–30 exhibiting reduced pain compared to normal-weight patients (coefficient = −6.66; *p* = 0.026).

### 3.10. Predictors of Alpha Angle

HU quartiles did not significantly predict the alpha angle (Appendix A). All other factors did not predict alpha angle as well.

### 3.11. Predictors of Beta Angle

HU quartiles were not significant predictors of the beta angle (Table 7). However, the fracture level and BMI significantly predicted this outcome. Specifically, patients with BMI 30–35 exhibited a higher beta angle compared to normal-weight patients (coefficient = 10.02; *p* = 0.045). Additionally, patients with thoracic fractures had significantly higher beta angles compared to those with lumbar fractures (coefficient = 4.92; *p* = 0.048).

### 3.12. Predictors of Complications

HU quartiles were not significant predictors of complications (Appendix A). All other factors did not predict complications as well.

### 3.13. Predictors of Death

HU quartiles were significant predictors of death (Table 8). Specifically, the second quartile, compared to the first quartile, showed a significantly higher risk of death (aOR = 8.12; *p* = 0.044) after accounting for potential confounders. Additionally, ASA grade III, compared to grade II, showed a lower risk of death (aOR = 0.164; *p* = 0.035).

## 4. Discussion

Osteoporotic fractures of the spine are a significant health concern, particularly among the elderly population [14]. These fractures often result from low-energy trauma and are associated with substantial morbidity, decreased quality of life, and increased mortality [15]. The thoracic and lumbar regions of the spine are particularly vulnerable to osteoporotic fractures due to their structural and functional roles. Effective preoperative planning is crucial for optimizing surgical outcomes and minimizing complications in patients with these fractures.

The results of this study provide valuable insights into the role of HU measurements obtained from preoperative CT scans in the management of osteoporotic fractures of the thoracic and lumbar spine. Contrary to our initial hypothesis, HU values did not significantly predict complications, length of hospital stay, or specific postoperative outcomes such as pain and spinal angles. However, HU quartiles significantly predicted mortality, with patients in the second quartile demonstrating a markedly increased risk compared to the lowest quartile (aOR = 8.12; *p* = 0.044). These findings suggest that HU values, particularly in lower ranges, may serve as an early indicator of mortality risk in this population.

### 4.1. Correlation with Quality of Life

The positive correlation between HU values and QoL suggests that patients with higher bone density, as indicated by higher HU values, tend to experience better postoperative quality of life. However, this observation in the updated regression analysis was deemed insignificant, although prior research indicates that better bone quality can lead to improved surgical outcomes and recovery [16].

### 4.2. Implications for Surgical Planning

While HU values did not predict complications or other specific outcomes, their association with QoL highlights the potential utility of HU measurements in preoperative planning. Surgeons could consider HU values when assessing patient prognosis and planning interventions that could enhance postoperative recovery and quality of life. This approach may be particularly beneficial in optimizing patient outcomes and ensuring targeted interventions for those at a higher risk of poor bone quality. This has been supported in the literature. Jeor et al. [17] demonstrated that lower HU values were independent predictors of osteoporosis-related complications (ORCs) such as proximal junctional kyphosis, pseudarthrosis, and instrument failure. The study found that, for every 25-point decrease in HU, the odds of an ORC increased by 1.7-fold, highlighting HU as a more reliable predictor than DXA T-scores. Zhang et al. [18] reported that HU values had the highest predictive efficacy for thoracolumbar fragility fractures compared to VBQ and DEXA-measured BMD. The area under the curve (AUC) for HU was 0.863, suggesting that HU measurements are robust indicators of fracture risk. Nguyen et al. [19] identified HU values, along with age and BMI, as significant predictors of worsening vertebral compression fractures (VCFs) following traumatic injury. The study’s regression model indicated that HU values are crucial for determining clinical follow-up and the need for surgical intervention.

Meanwhile, Jiang et al. [5] highlighted that low HU measurements of adjacent vertebrae or a high HU ratio between the surgical and adjacent vertebrae were significant risk factors for adjacent vertebral fractures post-PKP surgery. The study achieved high prediction accuracy, suggesting that HU measurements are effective in identifying patients at risk for subsequent fractures. Similarly, Ye et al. [20] found that lower vertebral HU values significantly increased the likelihood of new osteoporotic vertebral compression fractures (OVCFs) after PKP treatment. Both axial and sagittal HU evaluations showed high specificity and sensitivity in predicting new fractures, reinforcing the importance of HU as a predictor.

### 4.3. Clinical Outcomes: HU Quartiles and Other Predictors

Our analysis showed no significant differences in complications, length of hospital stay, or pain outcomes across different HU quartiles. This suggests that, while HU values reflect bone density, they may not be robust predictors of these specific clinical outcomes in the context of osteoporotic spinal fractures [21]. Factors such as the fracture level, Pfirmann classification, and BMI categories did not show significant predictive value either, further indicating the complex nature of surgical outcomes in this patient population. This aligns with previous research [22,23].

Interestingly, HU quartiles did not predict complications but significantly predicted mortality. The increased risk of mortality in the second quartile aligns with previous evidence, highlighting the role of bone density in overall patient outcomes. Furthermore, ASA category III emerged as a protective factor against mortality compared to ASA II (aOR = 0.164; *p* = 0.035), underscoring the importance of overall health status in patient survival. This finding aligns with the literature [24]. These findings underscore the importance of considering general health status alongside HU values in preoperative risk stratification. Notably, ASA category III was a significant negative predictor of mortality, underscoring the importance of overall health status in patient survival. This finding aligns with the literature [24].

While BMI categories were not significant predictors of most outcomes, higher BMI was associated with increased beta angles (30–35 category, coefficient = 10.02; *p* = 0.045) and reduced leg pain (25–30 category, coefficient = −6.66; *p* = 0.026). These findings suggest that BMI, as a surrogate for overall patient health, may influence specific surgical outcomes, particularly spinal alignment and postoperative comfort.

The observation that all patients with screw loosening or adjacent fractures had HU values below 65, despite not reaching statistical significance, suggests the clinical importance of considering bone quality in preoperative evaluations. The relationship between HU and screw loosening/dislodgement arises from the foundational biomechanical principle that bone quality determines its capacity to anchor hardware securely and resist stress. In patients with HU < 65, the combination of reduced anchorage strength and increased fragility might explain the higher prevalence of these complications. Integrating HU into risk stratification and surgical planning can help address this vulnerability and improve outcomes. Surgeons should integrate HU measurements into their decision-making process, as patients with low HU values may require tailored surgical strategies, such as augmented fixation techniques or bone density optimization preoperatively. This approach could potentially improve patient outcomes in cases of compromised bone quality.

### 4.4. Clinical Implications

Despite the absence of significant associations for complications and quality of life, the significant association of HU with mortality underscores its relevance in preoperative planning. Patients with low HU values may require additional preoperative interventions, such as augmented fixation strategies or medical optimization, to reduce their mortality risk. Moreover, the lack of predictive value for other outcomes, such as pain scores and length of hospital stay, highlights the complexity of factors influencing these variables and the need for larger datasets to capture potential associations.

### 4.5. Limitations and Future Research

This study has several limitations that warrant consideration. First, the small sample size limits the generalizability of our findings and reduces the statistical power of the analyses. While multivariate regression was performed to adjust for confounders, the small sample size restricted the inclusion of additional variables, such as age and frailty, which could provide valuable insights into clinical outcomes. Missing HU data for some patients further limited the scope of analysis. The available dataset contained only limited demographic data, with age at death being the sole available variable for a subset of patients (n = 20; mean = 81.62 years, SD = 6.77). Due to the scarcity of data, we were unable to incorporate age as a factor in the regression models, as the models yielded ’no observation’ when age was included. Furthermore, critical variables such as age at the time of presentation or treatment, which could provide valuable insights into its role as a risk factor or determinant, were not available in the dataset. Additionally, the high mortality rate in our study cohort underscores the importance of evaluating baseline frailty and age as potential covariates. However, frailty data were unavailable, and while ASA classification was used as a proxy for general health status, it does not fully capture the concept of frailty.

Second, variability in surgical techniques and fracture types introduces potential heterogeneity that could influence the outcomes. Although we attempted to address this by excluding certain therapy types due to collinearity and categorizing fractures into broader regions, residual variability may still affect the findings. Furthermore, this study did not include laboratory test data characteristic of osteoporosis, such as serum calcium, vitamin D levels, or bone turnover markers. The absence of such data limited our ability to investigate potential correlations between HU values and laboratory indicators of bone metabolism.

Third, the definitions of complications used in this study were broad and may not capture nuanced differences in complication severity. Additionally, long-term outcomes, which could provide deeper insights into the predictive value of HU, were unavailable. The lack of comprehensive follow-up data limits our ability to evaluate the full clinical trajectory of these patients.

Moreover, the retrospective design may introduce selection bias. Specifically, the absence of blinding in HU measurement and the evaluation of postoperative outcomes may introduce bias. Future prospective studies with larger, multicenter cohorts are necessary to validate these findings and explore the potential of HU measurements as part of a multifaceted approach to preoperative planning and risk assessment.

Further research should also investigate the integration of HU values with other imaging and clinical parameters to develop comprehensive predictive models for surgical outcomes. Understanding the interplay between bone quality, surgical technique, and patient-specific factors will be crucial in refining strategies to improve patient care.

## 5. Conclusions

Our findings demonstrate that, while HU values are not significant predictors of complications, pain scores, or quality of life, they are strongly associated with mortality risk. Patients in the second HU quartile exhibited an increased mortality risk, emphasizing the importance of HU in preoperative assessments. These results suggest that integrating HU values into routine preoperative evaluations could enhance risk stratification and guide surgical decision making for patients with osteoporotic spinal fractures. Future research should aim to validate these findings in larger, multicenter cohorts and explore the interplay between HU values, surgical techniques, and broader patient factors.

## Figures and Tables

**Table 1 medicina-61-00227-t001:** Baseline characteristics of included patients being treated surgically for osteoporotic vertebral fracture.

	Frequency/Mean	Percent/SD
**Age**	81.62	6.77
**Missing**	50	71.42
**Fracture level**		
Lumbar	35	50
Thoracic	30	42.86
Thoracolumbar	5	7.14
**Pfirmann classification**		
II	15	21.74
III	21	30.43
IV	19	27.54
V	14	20.29
**Therapy (MISS/LISS)**		
L1–L3	5	6.67
L2–L5	1	1.33
L2–L4	7	9.33
L3–5	3	4
Th11–L1	18	24
Th12–L2	19	25.33
Th10–12	5	6.67
Th12–L4	3	4
Th12–L1	2	2.67
Th8–10	1	1.33
Th16–8	1	1.33
Th5–6–8–10	1	1.33
Th5–10	1	1.33
Th8–12	1	1.33
Th10–L5	1	1.33
Th2–6	1	1.33
Th10–L2	1	1.33
Th7–9	3	4
Th11–L2	1	1.33
**BMI**		
<20	6	8.11
20–25	24	32.43
25–30	29	39.19
30–35	13	17.57
35–40	2	2.7
**ASA category**		
I	1	1.33
II	23	30.67
III	49	65.33
IV	2	2.67
**Hounsfield Unit**	*n* = 70	
	73.28	24.07
**Alpha angle (preop)**	*n* = 26	
	13.66	6.53
**Beta angle (postop)**	*n* = 26	
	16.27	7.45

SD: standard deviation; ASA: American Society of Anesthesiologists; BMI: body mass index; n: number of observations; preop: preoperative; postop: postoperative.

**Table 2 medicina-61-00227-t002:** A summary of the clinical outcomes of examined patients.

	Frequency/Mean	Percent/SD
**Blood loss (mL)**		
<100	35	46.67
100–500	38	50.67
500–1000	1	1.33
>1000	1	1.33
**Total complications**	36	52.17
Adjacent fracture	4	5.8
Liver cirrhosis	1	1.45
Respiratory insufficiency	1	1.45
MOF	2	2.9
CRS	1	1.45
Septic shock	3	4.35
Hemorrhagic shock	1	1.45
Hypokalemia	5	7.25
ICH	1	1.45
Spinal fusion	3	4.35
Pneumonia	6	8.7
Pneumothorax	2	2.9
Pleural effusion	1	1.45
Screw loosening	1	1.45
Screw migration	1	1.45
Screw tear-out	1	1.45
UTI	4	5.8
Tracheal stenosis	1	1.45
ARF	3	4.35
Hyponatremia	1	1.45
**Revision surgery**	1	1.45
**Death**	19	26.76
** Respiratory insufficiency**	1	5.26
** Multiorgan failure**	1	5.26
** Decompensated liver cirrhosis**	1	5.26
** Extended spinal fusion/screw tear-out**	2	10.53
** Cardiac decompensation**	1	5.26
** Cardio-renal syndrome**	1	5.26
** Septic shock**	1	5.26
** Hemorrhagic shock**	1	5.26
** Hypokalemia**	1	5.26
** Intracerebral hemorrhage**	1	5.26
** Adjacent fracture**	1	5.26
** Pneumonia**	2	10.53
** Tachyarrhythmia**	1	5.26
** Unclear/missing**	5	26.32
**Surgery duration (Std.)**		
<1 Std.	14	18.67
1–2 Std.	54	72
2 Std.	1	1.33
2–3 Std.	4	5.33
3–4 Std.	2	2.67
**LOS**	*n* = 75	
	13.38	7.2
**QoL (EQ-5D)**	*n* = 26	
	0.57	0.34
**VAS Back**	*n* = 26	
	3.88	2.38
**VAS Leg**	*n* = 26	
	2.5	2.91
**COMI**	*n* = 26	
	4.71	2.65
**Alpha angle (postop)**	*n* = 26	
	8.24	5.05
**Alpha angle (3 mo)**	*n* = 11	
	9.17	3.68
**Alpha angle (2 yr)**	*n* = 26	
	9.44	4.48
**Beta angle (postop)**	*n* = 26	
	10.74	5.56
**Beta angle (3 mo)**	*n* = 11	
	13.16	5.65
**Beta angle (2 yr)**	*n* = 26	
	14.09	6.25

LOS: length of hospital stay; mo: month; yr: year; SD: standard deviation; n: number of observations; COMI: core outcomes measures index; VAS: visual analog scale; QoL: quality of life.

**Table 3 medicina-61-00227-t003:** Correlation analysis between preoperative Hounsfield Units and clinical outcomes.

	HU	Alpha (Postop)	Alpha (3 mo)	Alpha (2 yr)	Beta (Postop)	Beta (3 mo)	Beta (2 yr)	COMI	QoL	LOS	VAS Back	VAS Leg	Surgery Duration
**HU**	1												
**Alpha (postop)**	0.0344	1											
*p*-value	0.8674												
**Alpha (3 mo)**	−0.3957	0.7811	1										
*p*-value	0.2284	**0.0045**											
**Alpha (2 yr)**	0.0649	0.5203	0.7691	1									
*p*-value	0.7529	**0.0064**	**0.0057**										
**Beta (postop)**	−0.1475	0.6236	0.6059	0.4692	1								
*p*-value	0.4721	**0.0007**	**0.0482**	**0.0156**									
**Beta (3 mo)**	−0.0615	0.4854	0.4622	0.0535	0.7752	1							
*p*-value	0.8576	0.1301	0.1523	0.8759	**0.0051**								
**Beta (2 yr)**	0.1339	0.154	0.5975	0.181	0.3022	0.8785	1						
*p*-value	0.5143	0.4525	0.0522	0.3762	0.1335	**0.0004**							
**COMI**	−0.3192	0.2233	−0.0846	−0.0269	0.2795	−0.0638	−0.3035	1					
*p*-value	0.112	0.2729	0.8047	0.8963	0.1667	0.8522	0.1317						
**QoL**	0.3362	−0.0957	0.2051	−0.0095	−0.057	0.4002	0.3763	−0.6292	1				
*p*-value	0.0931	0.642	0.5453	0.9631	0.7819	0.2226	0.0581	**0.0006**					
**LOS**	−0.1402	−0.0587	0.2843	−0.1575	0.1021	0.4598	0.4369	−0.1567	0.1049	1			
*p*-value	0.2472	0.7756	0.3968	0.4421	0.6198	0.1547	**0.0256**	0.4446	0.61				
**VAS Back**	−0.2364	0.4058	0.1923	0.0965	0.2567	0.3395	−0.0602	0.5807	−0.3281	−0.0932	1		
*p*-value	0.245	**0.0397**	0.5712	0.6392	0.2056	0.307	0.7702	**0.0019**	0.1018	0.6508			
**VAS Leg**	−0.1115	−0.0688	0.0851	−0.1533	−0.313	−0.2161	−0.3695	0.6362	−0.4522	−0.2542	0.3532	1	
*p*-value	0.5875	0.7386	0.8036	0.4548	0.1195	0.5234	0.0632	**0.0005**	**0.0204**	0.2101	0.0767		
**Surgery duration**	0.0201	−0.1582	−0.232	−0.4893	−0.231	0.1365	0.1308	−0.0957	0.2131	0.0752	0.0789	0.097	1
*p*-value	0.8687	0.4403	0.4924	**0.0112**	0.2562	0.689	0.5241	0.6419	0.2959	0.5214	0.7015	0.6373	

LOS: length of hospital stay; QoL: quality of life; mo: month; yr: year; VAS: visual analog scale; HU: Hounsfield Units; COMI: core outcomes measures index. The bolded values indicate statistically significant correlations (*p* < 0.05).

**Table 4 medicina-61-00227-t004:** Stratification of clinical outcomes based on Hounsfield Unit quartiles.

Variable	Category	Q1	Q2	Q3	Q4	Total	*p*
**Categorical Outcomes (Chi-Square Test)**							
**Blood loss**	<100	10 (30.3%)	4 (12.12%)	14 (42.42%)	5 (15.15%)	33	0.022
	100–500	7 (20%)	13 (37.14%)	4 (11.43%)	11 (31.43%)	35	
	500–1000	1 (100%)	0 (0%)	0 (0%)	0 (0%)	1	
	>1000	0 (0%)	0 (0%)	0 (0%)	1 (100%)	1	
**Complications**	No	7 (21.88%)	7 (21.88%)	7 (21.88%)	11 (34.38%)	32	0.181
	Yes	10 (29.41%)	9 (26.47%)	11 (32.35%)	4 (11.76%)	34	
**Death**	No	15 (30.61%)	11 (22.45%)	13 (26.53%)	10 (20.41%)	49	0.613
	Yes	3 (15.79%)	6 (31.58%)	5 (26.32%)	5 (26.32%)	19	
**Continuous Outcomes (ANOVA Analysis)**							
		**SS**	**df**	**MS**	**F**	** *p* **	**N**
**LOS**	Between groups	46.22428	3	15.40809	0.27	0.8442	69
	Within groups	3717.261	66	56.32214			
**COMI**	Between groups	27.82993	3	9.276644	1.37	0.2781	25
	Within groups	149.0129	22	6.773312			
**VAS Back**	Between groups	13.28242	3	4.427473	0.75	0.5324	25
	Within groups	129.3714	22	5.880519			
**VAS Leg**	Between groups	20.84286	3	6.947619	0.8	0.5084	25
	Within groups	191.6571	22	8.711688			
**QoL**	Between groups	0.99292	3	0.330973	3.87	0.0231	25
	Within groups	1.881909	22	0.085541			
**Alpha angle (postop)**	Between groups	49.86903	3	16.62301	0.62	0.6089	25
	Within groups	588.95	22	26.77045			
**Alpha angle (3 months)**	Between groups	15.41263	3	5.137544	0.3	0.8249	10
	Within groups	120.0187	7	17.14553			
**Alpha angle (2 years)**	Between groups	12.81497	3	4.271656	0.19	0.9006	25
	Within groups	489.1121	22	22.23237			
**Beta angle (postop)**	Between groups	64.94893	3	21.64964	0.67	0.579	25
	Within groups	709.9066	22	32.26848			
**Beta angle (3 months)**	Between groups	25.89436	3	8.631453	0.21	0.8894	10
	Within groups	293.8602	7	41.98003			
**Beta angle (2 years)**	Between groups	206.4728	3	68.82427	1.96	0.1496	25
	Within groups	772.7122	22	35.12328			

N: number of observations; df: degree of freedom; *p*: *p*-value; QoL: quality of life; VAS: visual analog scale; LOS: length of hospital stay; SS: sum of squares; MS: mean square; F: F-statistic; Q1: <56.24; Q2: 56.24–72.63; Q3: 72.63–87.59; Q4: >87.59.

**Table 5 medicina-61-00227-t005:** The multivariate linear regression model showing the predictors of length of hospital stay.

	Coefficient	SE	T	*p*	Low CI	High CI
**HU quartile [reference group: Q1 < 56.24]**						
**Q2: 56.24–72.63**	−0.939	2.923	−0.320	0.749	−6.816	4.937
**Q3: 72.63–87.59**	−0.468	2.738	−0.170	0.865	−5.975	5.038
**Q4 > 87.59**	−1.673	2.986	−0.560	0.578	−7.677	4.331
**Fracture level [reference group: lumbar]**						
**Thoracic**	2.448	2.213	1.110	0.274	−2.000	6.897
**Thoracolumbar**	−0.357	3.982	−0.090	0.929	−8.363	7.649
**Pfirmann classification [reference group: II]**						
**III**	−0.051	2.800	−0.020	0.985	−5.681	5.578
**IV**	−0.457	2.909	−0.160	0.876	−6.306	5.392
**V**	1.341	3.050	0.440	0.662	−4.792	7.474
**BMI category [reference group: 20–25/normal weight]**						
**<20**	−6.104	4.465	−1.370	0.178	−15.083	2.874
**25–30**	−2.860	2.315	−1.240	0.223	−7.515	1.794
**30–35**	−2.171	2.923	−0.740	0.461	−8.049	3.707
**35–40**	12.960	5.929	2.190	**0.034**	1.039	24.882
**ASA category [reference group: II]**						
**I**	−7.404	8.376	−0.880	0.381	−24.245	9.436
**III**	−2.037	2.386	−0.850	0.398	−6.835	2.761
**IV**	0.840	9.325	0.090	0.929	−17.910	19.589
**Constant**	16.213	3.305	4.910	0.000	9.567	22.858

MISS/LISS therapy was excluded due to significant collinearity with fracture level. Patients’ age at death was excluded due to insufficient number of observations. SE: standard error; CI: confidence interval. The bolded values indicate statistically significant correlations (*p* < 0.05).

**Table 6 medicina-61-00227-t006:** The multivariate linear regression model showing the predictors of leg pain score.

	Coefficient	SE	T	*p*	Low CI	High CI
**HU Quartile [reference group: Q1 < 56.24]**						
**Q2: 56.24–72.63**	2.725	3.134	0.870	0.407	−4.365	9.814
**Q3: 72.63–87.59**	1.430	2.616	0.550	0.598	−4.488	7.349
**Q4 >87.59**	1.843	2.801	0.660	0.527	−4.493	8.180
**Fracture level [reference group: lumbar]**						
**Thoracic**	−1.138	1.374	−0.830	0.429	−4.248	1.971
**Pfirmann classification [reference group: II]**						
**III**	2.630	3.891	0.680	0.516	−6.173	11.432
**IV**	2.689	2.552	1.050	0.319	−3.084	8.462
**V**	3.928	3.051	1.290	0.230	−2.975	10.831
**BMI category [reference group: 20–25/normal weight]**						
**<20**	−4.516	4.183	−1.080	0.308	−13.980	4.947
**25–30**	−6.661	2.498	−2.670	**0.026**	−12.311	−1.010
**30–35**	−5.160	2.758	−1.870	0.094	−11.400	1.080
**ASA category [reference group: II]**						
**I**	−0.583	4.154	−0.140	0.892	−9.981	8.815
**III**	−0.787	2.790	−0.280	0.784	−7.097	5.523
**Constant**	4.968	3.142	1.580	0.148	−2.141	12.076

MISS/LISS therapy was excluded due to significant collinearity with fracture level. Patients’ age at death was excluded due to insufficient number of observations. SE: standard error; CI: confidence interval. The bolded values indicate statistically significant correlations (*p* < 0.05).

**Table 7 medicina-61-00227-t007:** The multivariate linear regression model showing the predictors of beta angle.

	Coefficient	SE	T	*p*	Low CI	High CI
**HU quartile [reference group: Q1 < 56.24]**						
**Q2: 56.24–72.63**	−1.866	4.900	−0.380	0.712	−12.951	9.218
**Q3: 72.63–87.59**	6.725	4.091	1.640	0.135	−2.529	15.978
**Q4 > 87.59**	−4.336	4.380	−0.990	0.348	−14.243	5.571
**Fracture level [reference group: lumbar]**						
**Thoracic**	4.924	2.149	2.290	**0.048**	0.062	9.785
**Pfirmann classification [reference group: II]**						
**III**	1.282	6.084	0.210	0.838	−12.481	15.045
**IV**	−1.761	3.990	−0.440	0.669	−10.788	7.265
**V**	−1.453	4.771	−0.300	0.768	−12.246	9.340
**BMI category [reference group: 20–25/normal weight]**						
**<20**	10.347	6.541	1.580	0.148	−4.450	25.144
**25–30**	3.280	3.906	0.840	0.423	−5.555	12.115
**30–35**	10.016	4.313	2.320	**0.045**	0.260	19.773
**ASA category [reference group: II]**						
**I**	7.662	6.496	1.180	0.268	−7.032	22.356
**III**	−3.433	4.361	−0.790	0.451	−13.300	6.433
**Constant**	9.102	4.913	1.850	0.097	−2.012	20.217

MISS/LISS therapy was excluded due to significant collinearity with fracture level. Patients’ age at death was excluded due to insufficient number of observations. SE: standard error; CI: confidence interval. The bolded values indicate statistically significant correlations (*p* < 0.05).

**Table 8 medicina-61-00227-t008:** The multivariate logistic regression model showing the predictors of death.

	aOR	SE	T	*p*	Low CI	High CI
**HU quartile [reference group: Q1 < 56.24]**						
**Q2: 56.24–72.63**	8.124	8.430	2.020	0.044	1.063	62.102
**Q3: 72.63–87.59**	1.168	1.137	0.160	0.873	0.173	7.874
**Q4 > 87.59**	6.137	6.629	1.680	0.093	0.739	50.977
**Fracture level [reference group: lumbar]**						
**Thoracic**	0.791	0.622	−0.300	0.766	0.169	3.693
**Thoracolumbar**	5.550	7.648	1.240	0.214	0.373	82.657
**Pfirmann classification [reference group: II]**						
**III**	1.998	1.826	0.760	0.449	0.333	11.977
**IV**	0.422	0.434	−0.840	0.402	0.056	3.170
**V**	0.552	0.623	−0.530	0.598	0.060	5.047
**BMI category [reference group: 20–25/normal weight]**						
**<20**	1.000	(empty)				
**25–30**	0.373	0.304	−1.210	0.226	0.075	1.843
**30–35**	0.585	0.560	−0.560	0.575	0.090	3.814
**35–40**	1.315	2.280	0.160	0.874	0.044	39.340
**ASA category [reference group: II]**						
**I**	1.000	(empty)				
**III**	0.164	0.141	−2.110	**0.035**	0.031	0.882
**IV**	1.000	(empty)				
**Constant**	0.884	1.030	−0.110	0.916	0.090	8.667

MISS/LISS therapy was excluded due to significant collinearity with fracture level. Patients’ age at death was excluded due to insufficient number of observations. aOR: adjusted odds ratio; SE: standard error; CI: confidence interval. The bolded values indicate statistically significant correlations (*p* < 0.05). The bolded values indicate statistically significant correlations (*p* < 0.05).

## Data Availability

The dataset analyzed in this research can be provided by the corresponding author upon reasonable request.

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
