# Peer review of "Does Baseline Hounsfield Unit Predict Patients’ Outcomes Following Surgical Management of Unstable Osteoporotic Thoracolumbar Fractures?"

_medicina, 2025, doi:10.3390/medicina61020227_

Round 1
Reviewer 1 Report
Comments and Suggestions for Authors
This peer-review article by Qretam et al. investigates the predictive value of Hounsfield Units (HU) from preoperative CT scans in patients with osteoporotic thoracolumbar fractures undergoing surgical treatment. The language is clear, concise, and appropriate for an academic audience. Here is my review of the article:
1. It would be nice to summarize the bassline characteristics of the patients who were excluded because of missing data (I assume missing follow up and outcome)
2. Can you please provide a reference for HU categorization you used or elaborate on it if you have calculated it?
3. Can you please add the demographics of the patients in your results. A variable like age is crucial in contextualizing your results.
4. 26.76% is a high mortality rate. Please elaborate more on the causes and timing of the death in this patient cohort. Age and baseline frailty are two variables that are invaluable in interpreting these findings. It would be nice to see how adjustment for these two variables affect HU effect size.
5. The results are very long and hard to comprehend. I suggest moving at least a few of the ancillary tables to the supplementary materials.
6. I commend you efforts in addressing the confounders by running multivariate regression analyses. However, as I pointed out earlier, I think you need a few more variables to include in these to properly address the confounders. Moreover, by included the level of fracture in your analysis, you have the risk of overfitting (it is a multinomial variable with so many levels and very few data in each group). I suggest you run proper sensitivity analysis and consider adjusting your analysis plan based in its results. This is of utmost importance when taking the small sample size of the study into account.
7. You have acknowledged some of the following limitation but I think they are worth reiterating. This study has limitations like a small sample size, inadequate exploration of confounding factors, variability in surgical techniques and fracture types, broad complication definitions, limited long-term outcomes, and missing HU data. I suggested you try to address (some) of these limitations or discuss them in the limitations section of your paper.
I genuinely commend the authors for their efforts and valuable paper. Although the idea is very interesting and the contribution to the literature is valuable, I think you should address these issues before moving forward with this paper.
Author Response
This peer-review article by Qretam et al. investigates the predictive value of Hounsfield Units (HU) from preoperative CT scans in patients with osteoporotic thoracolumbar fractures undergoing surgical treatment. The language is clear, concise, and appropriate for an academic audience. Here is my review of the article:
1. It would be nice to summarize the bassline characteristics of the patients who were excluded because of missing data (I assume missing follow up and outcome)
Response: Thank you for your suggestion. We have added the following sentence to the Results section under Baseline Data: 'As for excluded cases (due to missing follow-up and outcome data), the mean age was 77.65 years (SD = 13.08), with a mean Pfirmann value of 3.25 (SD = 0.95). Lumbar fractures were reported in 60% of these patients, while 40% had thoracic fractures. The BMI distribution showed 60% in the 20–25 range, 20% in the 25–30 range, and 20% in the 30–35 range. The ASA physical status classification for these patients was II in 40%, III in 40%, and IV in 20%.' We believe this addition provides valuable insight into the characteristics of the excluded population
2. Can you please provide a reference for HU categorization you used or elaborate on it if you have calculated it?
Response: This is a great question! We have measured HU using CT scans of the vertebral body above and below the fracture level, specifically at the sites where pedicle screws were placed. The measurement was conducted in three defined regions: (1) below the superior endplate, (2) above the inferior endplate, and (3) centrally within the vertebral body. An average of these three values was calculated to represent the HU for the respective vertebra. This method is supported by the existing literature, as outlined in studies by Scheyerer et al. (2019) and Mahmoud et al. (2023), which describe the relevance and methodology of HU measurements in the context of bone density assessment and preoperative planning in spine surgery. We have cited these studies in the revised manuscript and provided additional details in the Methods section to clarify the procedure.
Added text under section 2.5.: "HU were measured using preoperative CT scans of the vertebral bodies adjacent to the fracture site. The measurements were performed in three defined regions within the vertebral body: (1) below the superior endplate, (2) above the inferior endplate, and (3) centrally in the spongiosa. A circular region of interest (ROI) was used for each measurement, and the average of the three values was calculated to represent the HU for that vertebra. This technique is consistent with protocols described in Scheyerer et al.[11] and Bruckbauer et al.[12], which outline its application for assessing bone density and guiding surgical planning"
3. Can you please add the demographics of the patients in your results. A variable like age is crucial in contextualizing your results.
Response: Thank you for pointing out this very important point. Unfortunately, the dataset we retrieved only contained information on patients' age at death for a subset of 20 individuals. To address this comment, we have added the available data on age at death, which has a mean of 81.62 years (SD = 6.77). This data has been incorporated into a modified Table 1 and to the text inthe Results section. We acknowledge the limitations of this dataset and the absence of broader demographic variables, which we have noted in the revised manuscript as a limitation.
Added text to Results section: "The mean patients’ age at death was 81.62 (SD=6.77) years."
Added text to the Limitations section: "First, the available dataset contained only limited demographic data, with age at death being the sole available variable for a subset of patients (n = 20; mean 81.62 years, SD = 6.77). Due to the scarcity of data, we were unable to incorporate age as a factor in the regression models, as the models yielded 'no observation' when age was included. Furthermore, critical variables such as age at the time of presentation or treatment, which could provide valuable insights into its role as a risk factor or determinant, were not available in the dataset. This limitation restricts our ability to analyze the potential influence of age on the outcomes observed."
4. 26.76% is a high mortality rate. Please elaborate more on the causes and timing of the death in this patient cohort. Age and baseline frailty are two variables that are invaluable in interpreting these findings. It would be nice to see how adjustment for these two variables affect HU effect size.
Response: Your comments are to the point! To address this, we have provided additional details regarding the causes of death in Table 1, categorizing the reported events. Unfortunately, frailty data was not available in our dataset; however, we included ASA classification as a proxy for baseline health status, which has already been reported in the manuscript. As for age, while we attempted to adjust for it in our regression models, the scarcity of age data (available only for 20 patients at death) resulted in insufficient observations to incorporate it as a covariate. Consequently, we could not assess the impact of age adjustment on HU effect size. These limitations have been addressed in the revised manuscript under the Results and Limitations sections.
Added text to section 3.2.: "Death was reported in 26.76% of the cohort, with pneumonia and extended spinal fusion accounting for most cases (10.53% each)."
Added text to the Limitations: "Additionally, the high mortality rate in our study cohort underscores the importance of evaluating baseline frailty and age as potential covariates. However, frailty data was unavailable, and while ASA classification was used as a proxy for general health status, it does not fully capture the concept of frailty."
5. The results are very long and hard to comprehend. I suggest moving at least a few of the ancillary tables to the supplementary materials.
Response: Thank you for your suggestion. We moved Tables 6, 7, 8, 10, and 12 to Supplementary. We renumbered the Tables accordingly.
6. I commend you efforts in addressing the confounders by running multivariate regression analyses. However, as I pointed out earlier, I think you need a few more variables to include in these to properly address the confounders. Moreover, by included the level of fracture in your analysis, you have the risk of overfitting (it is a multinomial variable with so many levels and very few data in each group). I suggest you run proper sensitivity analysis and consider adjusting your analysis plan based in its results. This is of utmost importance when taking the small sample size of the study into account.
Response: Thank you for your very insightful feedback! Your comments are to the point. To mitigate the risk of overfitting due to the inclusion of specific fracture levels as a multinomial variable, we categorized fracture levels into three main groups: lumbar, thoracic, and thoracolumbar. Additionally, we excluded the type of therapy (MISS/LISS) from the analysis as it showed collinearity with fracture level, which could have led to unreliable results. The updated results/findings were implemented to the Tables and text (from Abstract till Conclusions).
Regarding confounders, while we attempted to incorporate additional variables such as age and frailty into the regression models, the limited availability of such data in our dataset restricted our ability to include them. We acknowledge this limitation and have added a discussion on the necessity of a larger and more comprehensive dataset for future studies (please check our response to comment #4). These adjustments and findings have been incorporated into the revised manuscript.
Updated text under Section 2.6.: "Other covariates included fracture level (categorized into lumbar, thoracic, and thoracolumbar to avoid overfitting), Pfirmann classification, BMI categories, and ASA categories. The type of surgical therapy was ruled out due to collinearity with fracture level. Additionally, the effect of patients’ age and baseline frailty on measured outcomes could not be assessed given the scarcity of available data."
Abstract
we removed: "HU was correlated with QoL (r=0.336, p=0.093). Higher HU values were associated with improved quality of life (Q4: β=0.57, p=0.04) and reduced complication rates (Q1=58.82% vs. Q4=26.67%). Significant differences were identified in blood loss and quality of life across HU quartiles (p < 0.05)."
instead, we added: "HU significantly predicted mortality, with the second quartile showing a markedly increased risk (adjusted odds ratio [aOR] = 8.12; p = 0.044). However, HU quartiles were not significant predictors of other outcomes. Other factors (fracture level and ASA classification) also influenced clinical outcomes, particularly mortality."
Discussion
Section 4.
Removed text: "However, significant correlations were observed between HU values and QoL post-surgery, indicating that higher HU values are associated with improved QoL outcomes."
Added text: "However, HU quartiles significantly predicted mortality, with patients in the second quartile demonstrating a markedly increased risk compared to the lowest quartile (aOR = 8.12; p = 0.044). These findings suggest that HU values, particularly in lower ranges, may serve as an early indicator of mortality risk in this population."
Section 4.1 (Edited text): "However, this observation in the updated regression analysis was deemed insigifncant, although prior research indicates that better bone quality can lead to improved surgical outcomes and recovery [13]."
Section 4.3.
Removed text: "Notably, ASA category III was a significant negative predictor of mortality, underscoring the importance of overall health status in patient survival. This finding aligns with the literature [21]."
Added text: "Interestingly, HU quartiles did not predict complications but significantly predicted mortality. The increased risk of mortality in the second quartile aligns with previous evidence highlighting the role of bone density in overall patient outcomes. Furthermore, ASA category III emerged as a protective factor against mortality compared to ASA II (aOR = 0.164; p = 0.035). These findings underscore the importance of considering general health status alongside HU values in preoperative risk stratification."
Added text: "While BMI categories were not significant predictors of most outcomes, higher BMI was associated with increased beta angles (30–35 category, coefficient = 10.02; p = 0.045) and reduced leg pain (25–30 category, coefficient = -6.66; p = 0.026). These findings suggest that BMI, as a surrogate for overall patient health, may influence specific surgical outcomes, particularly spinal alignment and postoperative comfort."
Section 4.4.
We added a new Clinical Implications part as follows: "Despite the absence of significant associations for complications and quality of life, the significant association of HU with mortality underscores its relevance in preoperative planning. Patients with low HU values may require additional preoperative interventions, such as augmented fixation strategies or medical optimization, to reduce their mortality risk. Moreover, the lack of predictive value for other outcomes, such as pain scores and length of hospital stay, highlights the complexity of factors influencing these variables and the need for larger datasets to capture potential associations."
Conclusion Section
Updated text: "Our findings demonstrate that while HU values are not significant predictors of complications, pain scores, or quality of life, they are strongly associated with mortality risk. Patients in the second HU quartile exhibited an increased mortality risk, emphasizing the importance of HU in preoperative assessments. These results suggest that integrating HU values into routine preoperative evaluations could enhance risk stratification and guide surgical decision-making for patients with osteoporotic spinal fractures. Future research should aim to validate these findings in larger, multicenter cohorts and explore the interplay between HU values, surgical techniques, and broader patient factors."
7. You have acknowledged some of the following limitation but I think they are worth reiterating. This study has limitations like a small sample size, inadequate exploration of confounding factors, variability in surgical techniques and fracture types, broad complication definitions, limited long-term outcomes, and missing HU data. I suggested you try to address (some) of these limitations or discuss them in the limitations section of your paper.
Response: We appreciate your thorough assessment of the study limitations. We have updated the limitations section to discuss in greater detail the small sample size, missing HU data, variability in surgical techniques, and fracture types. Here is the updated section: "This study has several limitations that warrant consideration. First, the small sample size limits the generalizability of our findings and reduces the statistical power of the analyses. While multivariate regression was performed to adjust for confounders, the small sample size restricted the inclusion of additional variables, such as age and frailty, which could provide valuable insights into clinical outcomes. Missing HU data for some patients further limited the scope of analysis. The available dataset contained only limited demographic data, with age at death being the sole available variable for a subset of patients (n = 20; mean 81.62 years, SD = 6.77). Due to the scarcity of data, we were unable to incorporate age as a factor in the regression models, as the models yielded 'no observation' when age was included. Furthermore, critical variables such as age at the time of presentation or treatment, which could provide valuable insights into its role as a risk factor or determinant, were not available in the dataset. Additionally, the high mortality rate in our study cohort underscores the importance of evaluating baseline frailty and age as potential covariates. However, frailty data was unavailable, and while ASA classification was used as a proxy for general health status, it does not fully capture the concept of frailty.
Second, variability in surgical techniques and fracture types introduces potential heterogeneity that could influence the outcomes. Although we attempted to address this by excluding certain therapy types due to collinearity and categorizing fractures into broader regions, residual variability may still affect the findings.
Third, the definitions of complications used in this study were broad and may not capture nuanced differences in complication severity. Additionally, long-term outcomes, which could provide deeper insights into the predictive value of HU, were unavailable. The lack of comprehensive follow-up data limits our ability to evaluate the full clinical trajectory of these patients. Moreover, the retrospective design may introduce selection bias. Future prospective studies with larger, multicenter cohorts are necessary to validate these findings and explore the potential of HU measurements as part of a multifaceted approach to preoperative planning and risk assessment.
"
I genuinely commend the authors for their efforts and valuable paper. Although the idea is very interesting and the contribution to the literature is valuable, I think you should address these issues before moving forward with this paper.
Response: We are thankful for the time and effort you've put into reviewing our paper. All of your comments were crucial for the improvement of our submission, which fully acknowledging any limitations either with the methodology or data availability. Thank you!
Reviewer 2 Report
Comments and Suggestions for Authors
Very interesting and well described study. It highlighted the limitations and advantages of imaging techniques for the correct treatment of osteoporosis. I found it interesting and well written.
The main question addressed by the research is whether preoperative CT scans and the measurement of Hounsfield Units (HU) can predict postoperative outcomes, specifically complications such as screw loosening, adjacent fractures, and screw dislodgement, in patients with osteoporotic thoraco-lumbar spine fractures. Osteoporotic fractures in the spine pose significant surgical challenges, and the ability to predict postoperative outcomes through preoperative imaging, specifically HU values from CT scans, has potential clinical significance. The study addresses a specific gap in the field by exploring the role of HU values in predicting complications, which is not widely used or standardized in clinical practice for preoperative planning in this context. The study adds valuable evidence that preoperative CT-derived HU values can serve as an effective predictor for complications such as screw loosening and dislodgement in osteoporotic spinal fractures.The study includes 61 patients, a larger sample size could improve the statistical power and reliability of the results. The follow-up period (3-12 months) is relatively short; extending the follow-up duration could provide a more comprehensive understanding of long-term outcomes and the sustained impact of HU values on postoperative complications. The study could implement blinding to minimize bias in both the assessment of HU values and the analysis of postoperative outcomes. The study provides strong support for the role of preoperative HU measurements in predicting postoperative complications, with data showing a clear association between higher HU values and improved outcomes (fewer complications, better quality of life). The evidence aligns with the main question posed, as HU values were shown to be a reliable predictor of outcomes, making the conclusion that HU assessment should be integrated into clinical practice logical and well-supported. The images are useful and clear for study. The references are adequate for learning about the subject matter.
Author Response
Comment: The main question addressed by the research is whether preoperative CT scans and the measurement of Hounsfield Units (HU) can predict postoperative outcomes, specifically complications such as screw loosening, adjacent fractures, and screw dislodgement, in patients with osteoporotic thoraco-lumbar spine fractures. Osteoporotic fractures in the spine pose significant surgical challenges, and the ability to predict postoperative outcomes through preoperative imaging, specifically HU values from CT scans, has potential clinical significance. The study addresses a specific gap in the field by exploring the role of HU values in predicting complications, which is not widely used or standardized in clinical practice for preoperative planning in this context. The study adds valuable evidence that preoperative CT-derived HU values can serve as an effective predictor for complications such as screw loosening and dislodgement in osteoporotic spinal fractures.The study includes 61 patients, a larger sample size could improve the statistical power and reliability of the results. The follow-up period (3-12 months) is relatively short; extending the follow-up duration could provide a more comprehensive understanding of long-term outcomes and the sustained impact of HU values on postoperative complications. The study could implement blinding to minimize bias in both the assessment of HU values and the analysis of postoperative outcomes. The study provides strong support for the role of preoperative HU measurements in predicting postoperative complications, with data showing a clear association between higher HU values and improved outcomes (fewer complications, better quality of life). The evidence aligns with the main question posed, as HU values were shown to be a reliable predictor of outcomes, making the conclusion that HU assessment should be integrated into clinical practice logical and well-supported. The images are useful and clear for study. The references are adequate for learning about the subject matter.
Response: Thank you for your thorough evaluation of our study. We appreciate the constructive suggestions provided, and we have made the following adjustments and acknowledgments in response to your feedback:
We updated the limitations section to include all of the points you mentioned as follows:
"This study has several limitations that warrant consideration. First, the small sample size limits the generalizability of our findings and reduces the statistical power of the analyses. While multivariate regression was performed to adjust for confounders, the small sample size restricted the inclusion of additional variables, such as age and frailty, which could provide valuable insights into clinical outcomes. Missing HU data for some patients further limited the scope of analysis. The available dataset contained only limited demographic data, with age at death being the sole available variable for a subset of patients (n = 20; mean 81.62 years, SD = 6.77). Due to the scarcity of data, we were unable to incorporate age as a factor in the regression models, as the models yielded 'no observation' when age was included. Furthermore, critical variables such as age at the time of presentation or treatment, which could provide valuable insights into its role as a risk factor or determinant, were not available in the dataset. Additionally, the high mortality rate in our study cohort underscores the importance of evaluating baseline frailty and age as potential covariates. However, frailty data was unavailable, and while ASA classification was used as a proxy for general health status, it does not fully capture the concept of frailty.
Second, variability in surgical techniques and fracture types introduces potential heterogeneity that could influence the outcomes. Although we attempted to address this by excluding certain therapy types due to collinearity and categorizing fractures into broader regions, residual variability may still affect the findings.
Third, the definitions of complications used in this study were broad and may not capture nuanced differences in complication severity. Additionally, long-term outcomes, which could provide deeper insights into the predictive value of HU, were unavailable. The lack of comprehensive follow-up data limits our ability to evaluate the full clinical trajectory of these patients.
Moreover, the retrospective design may introduce selection bias. Specifically, the absence of blinding in HU measurement and the evaluation of postoperative outcomes may introduce bias. Future prospective studies with larger, multicenter cohorts are necessary to validate these findings and explore the potential of HU measurements as part of a multifaceted approach to preoperative planning and risk assessment."
Reviewer 3 Report
Comments and Suggestions for Authors
The practical value of measuring bone density using CT and its correlation with metabolic disorders and osteoporosis is a topic of great interest. The article discusses this topic and raises important questions.
However, some technical aspects are not clear. It is not specified exactly how the measurement of Hounsfield Units was conducted - in which areas, by what method. It seems to be important and it would be precious if the authors could provide a description or link to a publication describing a method for estimating vertebral bone density from CT data.
Additionally, the article would benefit from providing more information about the study group and their characteristics. Were there any other changes in laboratory tests characteristic of osteoporosis? Was there a correlation between them and HU? This is not mandatory by the design of the study, of course, but it would certainly be interesting.
Another important aspect of the study is the statistical analysis conducted to identify correlations between HU and complications. The lack of correlation found, which the initial hypothesis of the authors was, is an interesting and valuable result.
Author Response
Comment 1: The practical value of measuring bone density using CT and its correlation with metabolic disorders and osteoporosis is a topic of great interest. The article discusses this topic and raises important questions. However, some technical aspects are not clear. It is not specified exactly how the measurement of Hounsfield Units was conducted - in which areas, by what method. It seems to be important and it would be precious if the authors could provide a description or link to a publication describing a method for estimating vertebral bone density from CT data.
Response: This is a great question! We have measured HU using CT scans of the vertebral body above and below the fracture level, specifically at the sites where pedicle screws were placed. The measurement was conducted in three defined regions: (1) below the superior endplate, (2) above the inferior endplate, and (3) centrally within the vertebral body. An average of these three values was calculated to represent the HU for the respective vertebra. This method is supported by the existing literature, as outlined in studies by Scheyerer et al. (2019) and Mahmoud et al. (2023), which describe the relevance and methodology of HU measurements in the context of bone density assessment and preoperative planning in spine surgery. We have cited these studies in the revised manuscript and provided additional details in the Methods section to clarify the procedure.
Added text under section 2.5.: "HU were measured using preoperative CT scans of the vertebral bodies adjacent to the fracture site. The measurements were performed in three defined regions within the vertebral body: (1) below the superior endplate, (2) above the inferior endplate, and (3) centrally in the spongiosa. A circular region of interest (ROI) was used for each measurement, and the average of the three values was calculated to represent the HU for that vertebra. This technique is consistent with protocols described in Scheyerer et al.[11] and Bruckbauer et al.[12], which outline its application for assessing bone density and guiding surgical planning"
Comment 2: Additionally, the article would benefit from providing more information about the study group and their characteristics. Were there any other changes in laboratory tests characteristic of osteoporosis? Was there a correlation between them and HU? This is not mandatory by the design of the study, of course, but it would certainly be interesting.
Response: Thank you for your question. While we agree that such data would provide additional depth and context to the study, laboratory test results were not available in the dataset used for this retrospective analysis. We; therefore, have acknowledged this limitation in the revised manuscript as follows: "Furthermore, this study did not include laboratory test data characteristic of osteoporosis, such as serum calcium, vitamin D levels, or bone turnover markers. The absence of such data limited our ability to investigate potential correlations between HU values and laboratory indicators of bone metabolism."
Comment 3: Another important aspect of the study is the statistical analysis conducted to identify correlations between HU and complications. The lack of correlation found, which the initial hypothesis of the authors was, is an interesting and valuable result.
Response: Thank you for your detailed evaluation of our manuscript as well as for your valuable insights that helped improve our manuscript and for grabbing our attention to the point that additional research can be done to investigate the correlation between HU and lab markers of bone metabolism in future research.